# Clustering Pseudo Language Family in Multilingual Translation Models with Fisher Information Matrix

**Xinyu Ma**    **Xuebo Liu**[*]    **Min Zhang**

Institute of Computing and Intelligence, Harbin Institute of Technology, Shenzhen, China
mxinyuma@gmail.com, {liuxuebo,zhangmin2021}@hit.edu.cn

## Abstract

In multilingual translation research, the comprehension and utilization of language families are of paramount importance. Nevertheless, clustering languages based solely on their ancestral families can yield suboptimal results due to variations in the datasets employed during the model's training phase. To mitigate this challenge, we introduce an innovative method that leverages the fisher information matrix (FIM) to cluster language families, anchored on the multilingual translation model's characteristics. We hypothesize that language pairs with similar effects on model parameters exhibit a considerable degree of linguistic congruence and should thus be grouped cohesively. This concept has led us to define pseudo language families. We provide an in-depth discussion regarding the inception and application of these pseudo language families. Empirical evaluations reveal that employing these pseudo language families enhances performance over conventional language families in adapting a multilingual translation model to unfamiliar language pairs. The proposed methodology may also be extended to scenarios requiring language similarity measurements. The source code and associated scripts can be accessed at https://github.com/ecoli-hit/PseudoFamily.

## 1 Introduction

Multilingual neural machine translation (MNMT) aims to construct a single model to translate multiple languages and has proven its effectiveness (Aharoni et al., 2019). The application of such models for low-resource languages has revealed that leveraging supplementary languages during the fine-tuning phase can yield results that surpass traditional methods (Lakew et al., 2018; Gu et al., 2018). Nevertheless, the practical challenge lies in the development of the optimal strategy to identify the most beneficial auxiliary languages that can bolster the translation of low-resource language pairs.

Current academic investigations in this domain generally fall into two distinct categories: Firstly, the integration of diverse prior knowledge, which incorporates the utilization of various forms of existing knowledge and language family classifications, as evidenced in research such as Bakker et al. (2009); Chen and Gerdes (2017). Secondly, the exploration and computation of language similarity center on examining language representations and their comparative resemblances (Tan et al., 2019; Oncevay et al., 2020; Maurya and Desarkar, 2022).

Conventional methodologies predominantly rely on supplemental resources or the alteration of model architecture to categorize language families. However, those approaches need more adaptability, especially when dealing with recently developed pre-trained large MNMT models, e.g., m2m100 (Fan et al., 2021), where the original training data and model architecture are inaccessible. When it comes to a new scenario this limitation significantly hampers the effectiveness of existing methodologies for language family clustering.

To address this gap, we introduce a novel approach to pseudo language family clustering, which inherently depends on the MNMT model itself. Our proposed methodology hypothesizes that language pairs exhibiting similarities in their impact on model parameters are likely to possess a high level of congruence and should, consequently, be grouped together. We employ the FIM to quantify such similarities between language pairs. We demonstrate the efficacy of our approach by enhancing low-resource translation with pseudo language family clustering, yielding superior results compared to traditional language family clustering.

Our main contributions are as follows:

- We present an innovative methodology for clustering language families uniquely designed to function without the need for data access or architectural modifications.

---

[*]Co-first and Corresponding Author

- We unveil three efficient methods anchored in the FIM to comprehensively compute similarities amongst language pairs.

- Empirical evaluations indicate that our clustered pseudo language families exhibit superior performance compared to conventional language families. Furthermore, our approach exhibits versatility, effortlessly adapting to scenarios that necessitate language clustering.

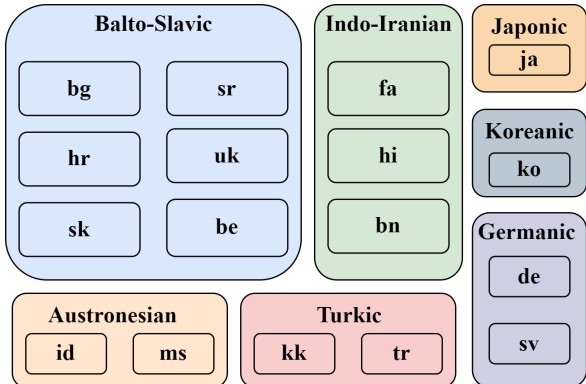

Figure 1: The used language families in our experiments.

## 2 Language Clustering with FIM

The accurate representation of language pairs is foundational for computing their pairwise similarities and ensuring efficacious clustering. As such, devising profound and robust representations for these pairs becomes imperative to attain excellence in this arena. Drawing inspiration from contemporary strides in parameter-efficient fine-tuning where innovative strategies have emerged to harness model parameters for representation (Ben Zaken et al., 2022; Gong et al., 2022; Zhang et al., 2023) we propose the deployment of the FIM. Our approach seeks to capitalize on the task-specific parameters embedded within the model.

### 2.1 Preliminary

**Dataset**  As an initial step, we curated a common language pool comprising translations of 17 languages to English (en), sourced from the TED Corpus (Qi et al., 2018). These language pairs collectively serve as the complete set for selection and function as the auxiliary language for low-resource languages. These languages span seven distinct families: Balto-Slavic, Austronesian, Indo-Iranian, Turkic, Japonic, Koreanic, and Germanic. The languages we employed are depicted in Figure 1.

**Model**  For empirical validation of our proposed methodologies, we elected to employ the m2m100_418M model (Fan et al., 2021). It is underpinned by a vast, many-to-many dataset, encompassing an impressive compilation of 100 languages. The innovation of this model lies in its unique data mining strategy, it capitalizes on linguistic similarities, thereby circumventing the need for exhaustive mining across all conceivable directions. Given the extensive breadth and innate multilingual capabilities of the model, it offers an apt platform for our investigative endeavors, furnishing a robust foundation for the appraisal and affirmation of our methodological propositions. For

consistency, we adhere to the model configuration and training settings delineated in Section 3.2.

**Fisher Information**  The concept of fisher information occupies a pivotal position in information theory, serving as an instrumental measure in discerning the importance and prospective value of tuning specific parameters. Essentially, it quantifies the variability of the first derivative of the likelihood function's logarithm. Through gauging the magnitude of this metric, one can deduce the necessity of fine-tuning particular parameters for subsequent tasks (Xu et al., 2021).

Formally, the FIM has been defined and employed in prior works (Theis et al., 2018; Thompson et al., 2019; Zhong et al., 2022) as:

$$
\begin{aligned}
\mathrm{F}_\theta = \mathbb{E}[(\nabla_\theta \log P(Y|X;\theta)) \\
(\nabla_\theta \log P(Y|X;\theta))^T]
\end{aligned}
\tag{1}
$$

Where $X$ and $Y$ denote the input and output respectively, and $\theta$ denotes the parameters of the model. For the $i$-th parameter in $\theta$ we empirically estimate its fisher information using the diagonal elements of the FIM. The formula derives as follows:

$$
\mathrm{F}_i = \mathbb{E}[(\nabla_i \log \mathrm{P}(Y|X;\theta))^2]
\tag{2}
$$

### 2.2 Calculating Similarity for Language Pairs

While leveraging a diagonal matrix aids in estimating FIM, arriving at precise probability estimations remains a non-trivial task. In light of this, we resort to the subsequent equation to approximate FIM:

$$
\mathrm{F}_i = \frac{1}{|\mathrm{D}|} \sum_j (\nabla_i \log \mathrm{P}(Y_j|X_j;\theta))^2
\tag{3}
$$

Here, D symbolizes the complete dataset, with $|\mathrm{D}|$ representing the data amount, and the entire dataset is divided into j mini-batches.

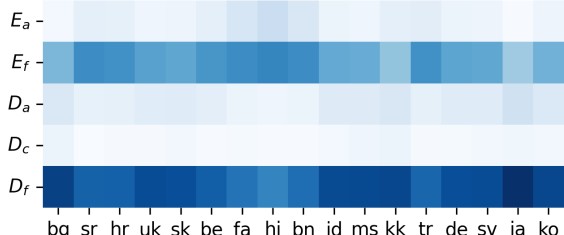

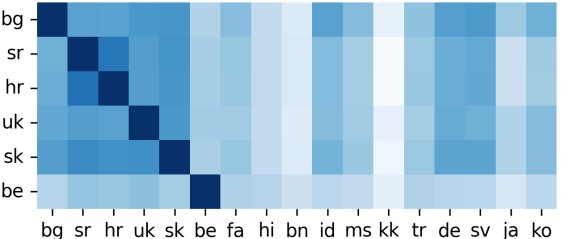

Figure 2: The parameters distribution of FIM for the training languages. We divide the model into five parts: encoder attention layer ($E_a$), encoder fully connected layer ($E_f$), decoder self-attention ($D_a$), decoder cross attention layer ($D_c$) and decoder fully connected layer ($D_f$).

We input parallel corpora corresponding to language pairs into the model and estimate the FIM using Equation 3 for just one epoch. We accumulate the FIM using the formula for each mini-batch during the model training process without backward propagation. After the completion of one epoch, we calculate the average FIM obtained from each mini-batch as the final estimation. Appendix A.1 provides the pseudo-code we use to estimate FIM.

**Significance of FFN in FIM Estimation** The model was then divided into five sections, and the distribution of the highest 40% parameters is depicted in Figure 2. It is pivotal to note that the feed-forward networks (FFN) contain over 60% of the selected parameters. A comparison was made with a related study (Meng et al., 2022) regarding the fine-tuning performance on the MNLI-m task.[1] In this study, the researchers solely fine-tuned a selected type of weight matrix from each transformer layer, and the best results were achieved when only the FFN was fine-tuned. From these observations, we conclude that the optimal strategy for this task involves utilizing the FIM derived from the FFN.

**Variants for Similarity Calculation** To effectively leverage the results derived from the FIM and judiciously select auxiliary languages for multilingual training of a specific target language pair, we devised three distinct strategies for calculating the similarity or distance between language pairs:

1. **Mean Square Error (MSE)**: Utilizing MSE offers an intuitive depiction of the distributional disparities among different language pairs. It is formally represented as:

---

Figure 3: The similarity computed with Overlap method, with darker colors representing the higher similarity between language pairs.

$$S_{(t,a)} = \frac{\sum (F_t - F_a)^2}{|F_t|} \quad (4)$$

Here, $t$ denotes the target language direction, and $a$ represents the auxiliary language pair. Furthermore, $F$ denotes the FIM, $F_a$ corresponds to the matrix associated with the auxiliary language, $F_t$ corresponds to the target language, and $|\cdot|$ symbolizes the matrix's size.

2. **KL Divergence (KL)**: By leveraging the FIM, we compute the KL divergence from other languages in relation to the target language, thereby capturing a more nuanced measure of the distances between language pairs.

$$S_{(t,a)} = |\sum F_a * \log(\frac{F_a}{F_t})| \quad (5)$$

The nomenclature used here mirrors that of the MSE, with $|\cdot|$ indicating absolute value.

3. **Overlap Similarity (Overlap)**: In further leveraging the FIM, we construct a fisher information mask. Here, the uppermost $K$ parameters are attributed a value of 1, whilst the others are designated a value of 0. The similarity amongst each language pair is ascertained using:

$$S_{(t,a)} = \frac{\text{Overlapping}(M_t, M_a)}{\text{Activate}(M_t)} \quad (6)$$

Here, $M$ symbolizes the fisher information mask, with Overlapping and Activate respectively quantifying the number of parameters that are simultaneously activated and those activated in the target direction.

### 2.3 Clustering Pseudo Language Family

**Selection of Pseudo Language Families** For the selection of auxiliary languages, we designed a

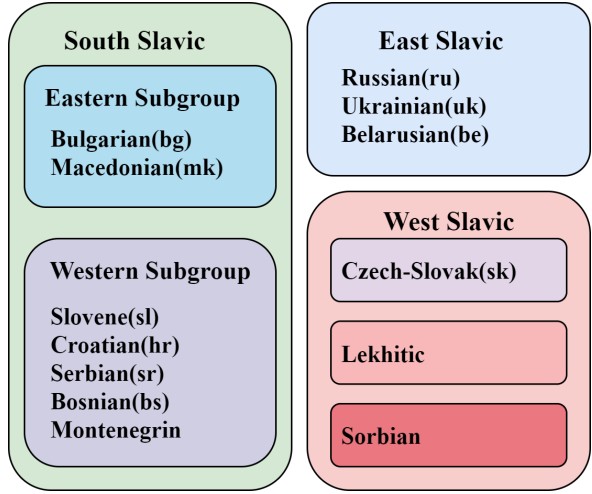

Figure 4: The phylogenetic tree of Slavic languages.

pragmatic algorithm. Beginning with the sorting of similarities between language pairs, we set an initial search radius. Within this pre-defined boundary, the language pairs in closest proximity are integrated into the auxiliary language roster. The radius is then adjusted, taking cues from the similarity metric of the latest added language pair. This process is iterated until the list ceases to expand with new language pairs. As a result, we identify a cluster that we term the pseudo language family.

This algorithm progressively broadens its selection boundary but at a decelerating rate, aiming to assimilate languages beneficial for the auxiliary set while sidestepping those exhibiting pronounced dissimilarities. For a comprehensive insight into the algorithm, refer to Appendix A.2.

**Analysis** We examined the efficacy of FIM in gauging linguistic similarity by juxtaposing it with the Slavic language family structure in Figure 4. This comparison was made using the similarity matrix derived from the Overlap method, as illustrated in Figure 3. The insights obtained from FIM are consistent with the findings of phylogenetic tree analysis. For instance, Serbian (sr) and Croatian (hr), situated on the same terminal branch, are anticipated to exhibit a higher similarity score.

In contrast, languages such as Ukrainian (uk), which belongs to the East Slavic linguistic group, are situated remotely on the phylogenetic tree. This geographical and linguistic distance results in only moderate similarity scores when compared with non-East Slavic languages in our matrix. These observations are aligned with the insights presented by Kudugunta et al. (2019).

## 3 Improving MNMT with Pseudo Family

### 3.1 Searching the Best *K* for Overlap Method

To ascertain this threshold, we embarked upon a series of experimental procedures with the range spanning from 30% to 80%. Our meticulous analyses culminated in the selection of 40% as the default parameter for the Overlap method, given its superior performance during the evaluation of the overall performance on the developmental set.

### 3.2 Main Experiments

**Model and Dataset** We employ the same m2m100_418M model as outlined in (Fan et al., 2021) as the foundational and utilize the datasets delineated in Section 2.1. For our experiments, we selected languages from the Indo-Iranian and Austronesian language families to test our methods, considering all language pairs as potential auxiliary languages. All the data has been cleaned and tokenized with fairseq script[2] and mosesdecoder[3].

**Baselines** For our evaluation, we considered the following baselines based on the primary model: **1) Vanilla**: Direct translations were conducted using the pre-trained model for the target language pairs without fine-tuning; **2) FT**: The primary model underwent a fine-tuning phase using bilingual data specific to the target language pairs; h**3) LF**: The primary model was fine-tuned by employing the traditional language family delineated in Figure 1, with temperature sampling and set temperature at 1.5; **4) LF+FT**: Expanding on the **LF** method, we integrated an additional fine-tuning phase that used the target language pair data.

**Training and Inference** For the training stage, we set the batch size as 4096; for our methods, we up-sample the training data to be the same to ensure that the proportion of each language within each mini-batch is equal except Hindi (hi), which is aligned with **LF** when using Overlap method. For optimization, we deployed the Adam optimizer (Kingma and Ba, 2015) with $\beta_1 = 0.9$, $\beta_2 = 0.98$, and $\epsilon = 10e^{-6}$. A learning rate of $lr = 3e^{-5}$ was chosen for this process.

For the inference stage, beam search was employed with a beam size set to 5 and a length penalty 1.0 applied consistently across languages. Evaluation metrics were based on BLEU scores,

---

[2]https://github.com/facebookresearch/fairseq/tree/main/examples/m2m_100

[3]https://github.com/moses-smt/mosesdecoder

| Model | fa | hi | bn | id | ms |
|---|---|---|---|---|---|
| **KL** | id sv de | bg de | sv uk id | bg sv | id bg sv |
| **MSE** | tr ko id | fa be tr | hi be fa | bg sv de | id bg ko sk |
| **Overlap** | tr ko id | bn fa | hi fa tr | bg ms sv | id bg sv de uk |

Table 1: The pseudo language family formed by the similarities computed using our methods.

calculated using SacreBLEU (Post, 2018). Our experimental framework was built upon the fairseq toolkit (Ott et al., 2019).

| Model | fa | hi | bn | id | ms | Avg. |
|---|---|---|---|---|---|---|
| | *Baselines* | | | | | |
| **Vanilla** | 25.0 | 19.5 | 10.7 | 28.0 | 28.6 | 22.4 |
| **FT** | 36.5 | 35.1 | 23.0 | 39.4 | 35.3 | 33.9 |
| **LF** | 33.9 | **35.6** | 26.3 | 38.4 | 32.5 | 33.3 |
| **LF+FT** | 36.8 | **35.6** | 26.3 | 39.0 | 32.5 | 34.0 |
| | *Our Methods* | | | | | |
| **KL** | 36.1 | 35.5 | 25.2 | **40.0** | **37.6** | 34.9 |
| **MSE** | **36.9** | 34.9 | 27.1 | 39.9 | 35.9 | 34.9 |
| **Overlap** | **36.9** | **35.6** | **27.2** | 39.7 | 35.5 | **35.0** |

Table 2: The main result of our experiment on baselines and our pseudo family method with the three different methods to calculate similarities. The threshold of $K$ in Overlap method is set as 40%.

## 3.3 Result and Analyse

**Comparison between Language Families** A deeper investigation into language families, as depicted in Table 1, reveals a critical insight: within the traditional structure of a language family, it is not guaranteed that all languages will provide superior translation outcomes, even precipitating a significant compromise in the translation quality.

Intriguingly, within the pseudo language family, our analysis identified language pairs that were linguistically divergent from the target language. For instance, Korean (ko) was selected as an auxiliary language for translation from Farsi (fa) to en. Data similarity metrics show that fa exhibits a relatively higher similarity with ko than with others, as we have detailed in Appendix A.3.

This phenomenon highlights that our methodology is proficient in tapping into dimensions beyond traditional linguistic boundaries, thereby selecting language pairs or datasets that are more conducive to optimal model training. We have delved deeper into the comparative analysis substantiating the superiority of our method in Appendix A.4.

**Effect of Pseudo Language Family** Our main results, comparing the outcomes between auxiliary languages identified through our proposed methods and the baselines, are showcased in Table 2. Our pseudo family strategies delivered notable performance uplifts across all three methods, clearly outpacing the conventional language family method.

Averagely, we noted a boost of 1.7 BLEU scores when juxtaposed against the traditional language family approach, a rise of 1.1 BLEU when compared to pure fine-tuning, and a 1.0 BLEU elevation against the hybrid approach of fine-tuning post multilingual training. A notable leap in performance was witnessed for several low-resourced language pairs, underpinning the efficacy of our strategy.

Furthermore, we assessed our methodology's resilience across diverse datasets. Our proposed method adeptly marries linguistic pertinence with data-related similarity, facilitating the astute selection of auxiliary languages for optimized translation outcomes. For comprehensive details, please refer to the Appendix A.5.

## 4 Conclusion and Future Work

In this study, we introduce a pioneering approach and three methods to quantify the similarities between language pairs and subsequently propose a pseudo family, based on this measurement, to enhance the selection of auxiliary languages for multilingual training in low-resource language scenarios. Our strategies for assessing language similarity leverage fisher information to sieve through model parameters, subsequently identifying task-specific ones and utilizing them to formulate an FIM to gauge the similarity between language pairs. The experimental outcomes significantly improve BLEU scores on low-resource languages, affirming our proposed method's feasibility and accuracy.

Future works include: 1) expanding the application of the proposed FIM methodology to other tasks that require the calculation of language similarity and 2) integrating our findings into curriculum learning frameworks (Liu et al., 2020; Zhan et al., 2021) and consistency learning frameworks (Li et al., 2022; Wang et al., 2022; Liu et al., 2023; Li et al., 2023).

## Limitations

In the present study, our methodology and experiments predominantly concentrated on the analysis of low-resource language pairs. An exploration of the generalization performance of our approach across a more comprehensive range of language pairs would be of significant value. Moreover, while our experiments employed the m2m100_418M model, future investigations could benefit from incorporating larger m2m100 models. Testing the generalization capacity of our method with diverse models, such as the NLLB (Costa-jussà et al., 2022) and SeamlessM4T (Barrault et al., 2023), could yield more valuable insights.

Furthermore, it is pertinent to note that the FIM retains extensive information. In forthcoming studies, we intend to implement innovative, parameter-efficient fine-tuning strategies akin to those propounded by (Gong et al., 2022). These strategies aim to ameliorate the detection of pertinent parameters while concurrently managing the total number of parameters with greater efficacy.

## Acknowledgment

This work was supported in part by the National Natural Science Foundation of China (Grant No. 62206076), Shenzhen College Stability Support Plan (Grant Nos. GXWD20220811173340003, GXWD20220817123150002), Shenzhen Science and Technology Program (Grant No. RCBS20221008093121053). Xuebo Liu was sponsored by CCF-Tencent Rhino-Bird Open Research Fund. The computational resources of this research were supported by the Education Center of Experiments and Innovations at Harbin Institute of Technology, Shenzhen. We would like to thank the anonymous reviewers and meta-reviewer for their insightful suggestions.

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

# A  Appendix

## A.1  Process of Similarity Calculation

---

**Algorithm 1:** FIM Estimation

**Data:** $\Theta$ : parameters weight; D: training data; |D|: number of mini-batches in dataset D; $L(\Theta)$: loss function for parameters $\Theta$; **sample**$_t$ : mini-batch for t-th step; **FIM**$_t$: fisher information matrix

1 **initialization**: Begin training for epoch one; $\Theta$ is inherent from the pre-trained model; **FIM**$_0 \leftarrow 0$; $t \leftarrow 0$;

2 **for** *sample*$_t \leftarrow$ D **do**

3    $L(\Theta) = step\_traning(\Theta)$;

4    // training for a mini-batch of dataset D;

5    $g_t = \frac{\partial L(\Theta)}{\partial \theta}$;

6    // Get gradients;

7    **FIM**$_{t+1} \leftarrow$ **FIM**$_t + \frac{g_t^2}{|D|}$;

8    $t \leftarrow t + 1$ ;

9    // update FIM;

10    // do not backpropagation;

11 **end**

12 return **FIM**$_t$

---

In order to identify pertinent language pairs for multilingual training, we advocate for the application of the FIM similarity methodology, which has demonstrated commendable efficacy. Our proposed approach can be delineated into three phases:

1. Construction of Shared Language Pool: A shared language pool, embodying a plethora of languages across diverse language families, is constructed.

2. Derivation of FIM: The fisher information matrix for each language in this pool is then computed using the estimation strategy detailed in Section 2.2, and the pseudo-code is shown in Algorithm 1.

3. Computation of Language Pair Similarity: When identifying suitable auxiliary languages for multilingual training with a designated target language pair, we employ three methods described in Section 2.2 to calculate the similarity between each language pair.

## A.2  Algorithm for Language Selection

The algorithmic details for the selection of auxiliary language pairs are as follows:

1. Sort the similarities in descending (ascending for MSE and KL) order to create a list $L$,

2. Initialize $Gap$ as $|L[1] - L[0]|$, add the first language into the auxiliary list,

3. Iterate from $i = 2$ to the end of $L$, for each loop we select language as follow:

   1) If $|L[i - 1] - L[i]| < Gap$ add the $i$-th language into the auxiliary list and update the $Gap = |L[i - 1] - L[i]|$

   2) If $|L[i - 1] - L[i]| = Gap$ add the $i$-th language into the auxiliary list and update the $Gap = \frac{Gap}{2}$;

   3) If $|L[i - 1] - L[i]| > Gap$, terminate the loop;

4. The language pairs in the auxiliary list with the target pair are composed of the pseudo language family for the target language pair.

## A.3  Data Similarity Analysis of Pseudo Family

From a linguistic vantage point, ko and fa languages demonstrate stark disparities. They neither share analogous characters nor exhibit proximal grammatical structures. Despite these disparities, it is pertinent to highlight the prevalence of word embedding as a method to abstractly represent characters and sentences, thereby bridging the gap between linguistically disparate languages. Through word embedding, characters or phrases with analogous meanings may find themselves in closer proximity within the embedded space.

Taking this into consideration, our analysis transcends mere linguistic similarities and delves into the dataset's similarities as well. We have employed a sampling methodology to ascertain the data similarity between fa and both traditional and pseudo language pairs:

1. Randomly select 10% data points from each auxiliary language training set, denoted as $T$.

2. Randomly select 10% data points from the training set of fa denoted as S.

3. For each data point in $T$, compute the cosine distance to all data points in $S$.

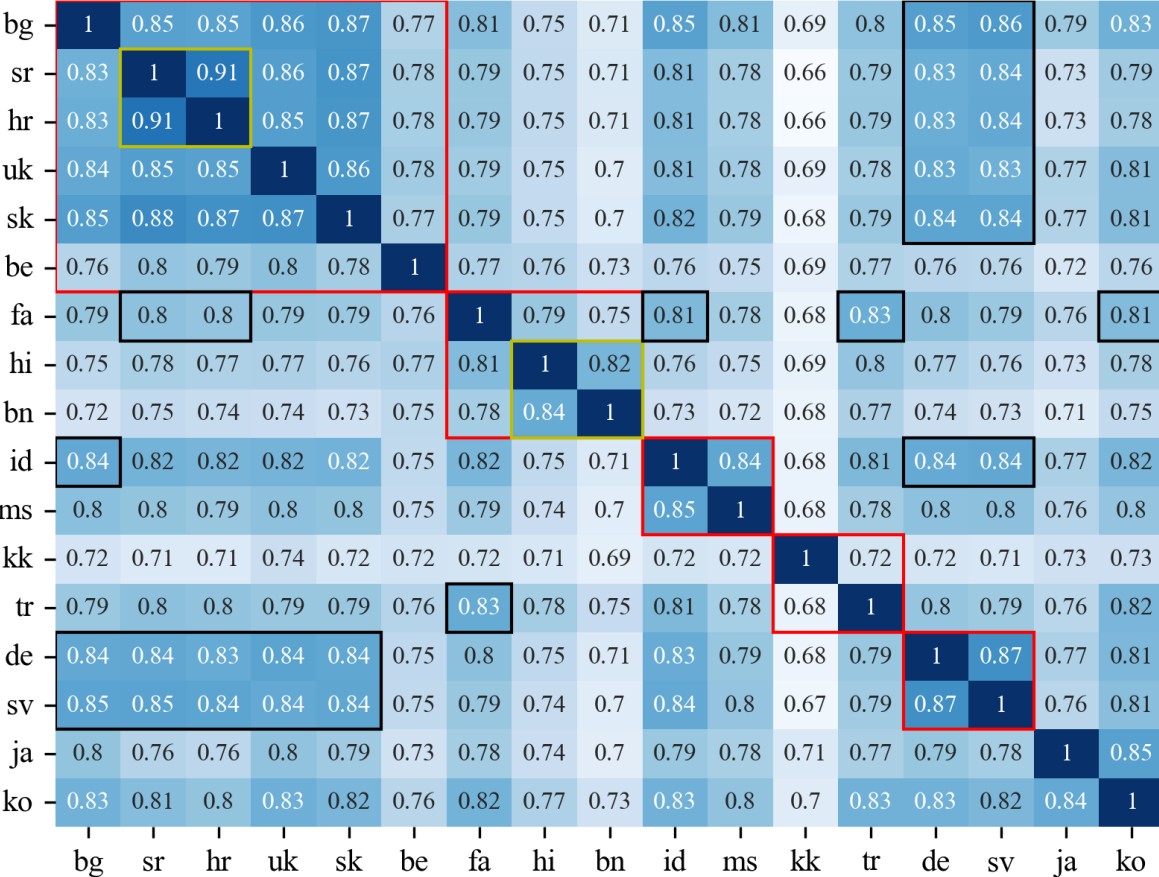

Figure 5: The full similarities computed TED datasets using Overlap method. We can directly infer the similarity between language pairs by observing the darkness or lightness of colors. Red boxes cover a language family, yellow boxes demonstrate the potential subfamily and black boxes contain some auxiliary languages beyond the language family that can leverage the target language-pair's translation.

4. For each data point in $T$, average the cosine distances of its top 5 closest data points in $S$ as similarity.

5. Finally, average the calculated similarities for all data points in $T$.

The conclusive outcomes demonstrate that within the dataset, ko and fa share a similarity of 32.6%, comparable to the likeness between Bengali (bn) at 32.6% and hi at 33.5%, with the latter duo forming part of the traditional language family of fa. Significantly, ko boasts a more substantial data volume in comparison to bn and hi, a factor that could considerably bolster the training procedure. This observation underscores that our methodology does not exclusively hinge upon linguistic similarities; instead, it amalgamates information gleaned from the dataset, a process that is facilitated by the model's representational form.

### A.4 Supplementary Analysis of Language Families

Figure 5 elucidates that one can discern subgroups within a vast language family, thereby alluding to the intricate web of language relationships and the potential for additional sub-categorizations. Categorizing sub-language families solely based on linguistic methodology can be a formidable task.

Furthermore, it is apparent that languages chosen due to their intrinsic mutual similarities may not necessarily be part of the same traditional language family. This phenomenon echoes our methodology, wherein the model's parameters serve as a conduit for representation, thereby facilitating the amalgamation of multifaceted information spanning languages and data, which could potentially unveil novel insights that could bolster our comprehension of pre-trained models.

Moreover, traditional methodologies often employ stringent classifications for language pairs,

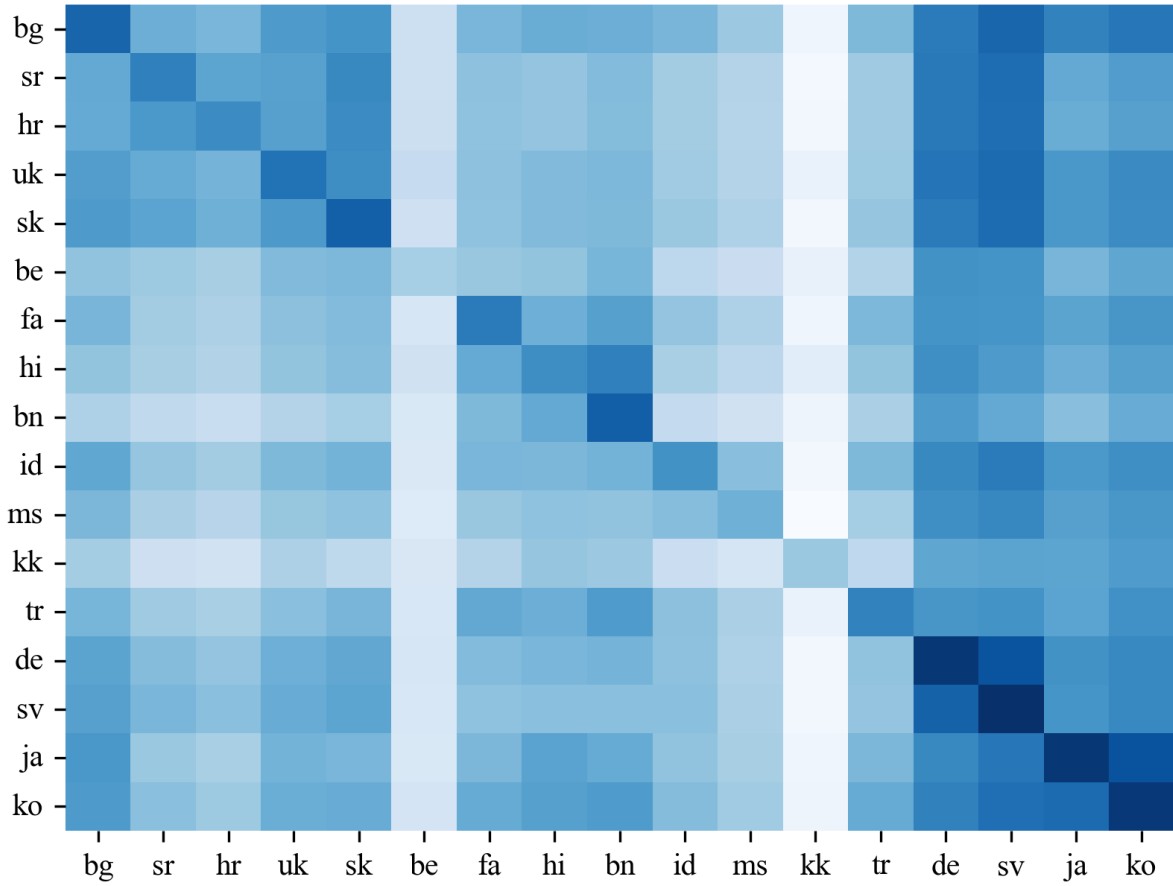

Figure 6: The full similarities computed with OPUS100 and TED datasets. Even when computing the similarity between language pairs across various datasets, most similarities remain highest for the same language pair within different datasets.

which may prove inadequate when navigating languages that hover near classification boundaries, potentially leading to incongruities. Additionally, such a rigid approach might fall short when faced with shifts in the data domain.

In contrast, our approach embraces a more fluid and dynamic selection algorithm that meticulously curates choices tailored to each target language pair. Crucially, our method is versatile, accommodating variations across different models and datasets, thereby ensuring its seamless applicability across a spectrum of scenarios. This versatility is highlighted by the observation that numerous languages can seamlessly integrate into multiple pseudo-language families.

### A.5 Robustness Analysis Across Different Datasets

To more comprehensively demonstrate the performance of our methodology across diverse domain-specific datasets, we conducted an in-depth analy-

sis to scrutinize the variations in similarities generated by our approach across diverse data combinations. For the purposes of our experimentation and analysis, we employed the OPUS100 dataset to facilitate our experimental and research endeavors. Subsequently, we calculated $S_{(t,a)}$ where $t$ is derived from the TED dataset and $a$ originates from the OPUS100 dataset. Figure 6 documents this full similarity matrix.

Notably, our findings demonstrate that even when languages are derived from distinct corpora, there remains a pronounced similarity among linguistically akin ones. This underscores the robustness of our methodology in discerning intrinsic linguistic affinities.