# OpenReview forum: "Clustering Pseudo Language Family in Multilingual Translation Models with Fisher Information Matrix"
_EMNLP/2023/Conference — EMNLP 2023 Main_

### Official Review · Reviewer_kKvr · 2023-08-02

**Soundness:** 3

**Excitement:**

3: Ambivalent: It has merits (e.g., it reports state-of-the-art results, the idea is nice), but there are key weaknesses (e.g., it describes incremental work), and it can significantly benefit from another round of revision. However, I won't object to accepting it if my co-reviewers champion it.

**Missing References:**

Give a reference to FIM already in the Introduction where it is mentioned first.

The given reference for the TED corpus on line 104 is wrong. Chronopoulou et al. use the TED corpus in a similar manner as the current paper, but for the TED talks the reference should be this one which is used by Chronopoulou et al.: https://aclanthology.org/N18-2084/

**Paper Topic And Main Contributions:**

The paper introduces a way to calculate translation pair similarity using Fisher information matrix (FIM) in order to create pseudo language-families. The families are used to choose auxiliary data in fine-tuning tranlastion models. The method is evaluated against using traditional language families.

**Questions For The Authors:**

Q A: If I understood correctly, you did not try to fine-tune each of the languages on all of the languages of the TED dataset?

Q B: In Figure 4, the similarity is always 1 if the language is itself. Would it still be 1, if the material would not be the same?

**Reasons To Accept:**

The idea of using FIM to calculate similarity and use that similarity to create new pseudo families is certainly an interesting one.

**Reasons To Reject:**

I'm not convinced that the presented method gives considerable enhancements and that its superiority is proven by the given experiments. It would need more evaluation. There are so many things here that can affect the end result. For example, why is ms so much better with FT than with others?

The paper ignores the fact that some of the languages with traditional language families are written with different writing systems. But then, so are they in the generated pseudo families in thia paper. I'm not convinced that Farsi model can learn anything from the Korean text which does not share single unicode character. Would the results be the same if just the English translations would be used to fine tune the models and there would not be any source text or the same sentence in Korean every time?

The paper mentions variations in the domain of the data utilized in the abstract, but it is not later investigated further. Does having different domain help or hinder the outcome? Are the results better because the model is seeing new English sentences and does not have that much to do with the source language?

**Reproducibility:**

4: Could mostly reproduce the results, but there may be some variation because of sample variance or minor variations in their interpretation of the protocol or method.

**Reviewer Confidence:**

2: Willing to defend my evaluation, but it is fairly likely that I missed some details, didn't understand some central points, or can't be sure about the novelty of the work.

**Typos Grammar Style And Presentation Improvements:**

Space missing before citation on lines 35 and 210.

In 2.1 on lines 110 and 111 it said that the respective volumes of the dataset would be presented on Table 1. They are not presented there.

In Tables 2 and 3 and in their captions: "Pesudo" -> "Pseudo".

---

> ### Author Rebuttal · Authors · 2023-08-29
>
> Thanks for the review.
>
> > Q1: I'm not convinced that the presented method gives considerable enhancements and that its superiority is proven by the given experiments. It would need more evaluation. There are so many things here that can affect the end result. For example, why is ms so much better with FT than with others?
>
> Your query is valid and we understand the need for clarity on this matter. A key aspect contributing to the performance difference is the similarity between the training and test data. When there's a high similarity between the two datasets, the FT model tends to outperform the LF. This is evident from the computed similarities for the language pairs we experimented with. However, the use of Fisher information in our pseudo LF method allows it to consider data similarity and other related factors at the same time, thereby providing results superior to both LF and FT.
>
> To calculate the similarity between training and test datasets, we used a sampling method:
> 1. Randomly select 1000 data points, denoted as T, from the training set.
> 2. Randomly select 100 data points, denoted as S, from the test set.
> 3. For each data point in T, compute the cosine distance to all data points in S.
> 4. For each data point in T, average the cosine distances of its top 5 closest data points in S.
> 5. Finally, average the calculated similarities for all data points in T.
>
> Table 1: The similarity between the training set and test set in each language pair.$ \Delta = BLEU_{FT} - BLEU_{Vanilla}$
>
> |            | fa                  | hi                  | ms                  |
> | ---------- | ------------------- | ------------------- | ------------------- |
> | similarity | 0.10810209134725653 | 0.12462727531056876 | 0.13408131563654796 |
> | $\Delta $  | +11.5               | +15.6               | +16.7               |
>
>
>
> > Q2: The paper ignores the fact that some of the languages with traditional language families are written with different writing systems. But then, so are they in the generated pseudo families in this paper. I'm not convinced that Farsi model can learn anything from the Korean text which does not share single unicode character. Would the results be the same if just the English translations would be used to fine tune the models and there would not be any source text or the same sentence in Korean every time?
>
> Your skepticism, rooted in linguistic differences, is understandable. However, when using multilingual pre-trained models, the characters and sentences are transformed into vector representations post the embedding layer. This abstract representation allows languages to aid one another in the learning process, regardless of script differences.
>
> To illustrate, we computed the distance between common words in Korean and Farsi using the vanilla M2M model. Specifically, Korean word embeddings were utilized to identify the nearest Farsi word via Euclidean distance.
>
> Table 2: Korean words and their closest Farsi counterparts based on embeddings.
>
> | ko     |           | fa    |               |
> | ------ | --------- | ----- | ------------- |
> | 만렙   | max level | تفریح | entertainment |
> | 마음   | mind      | ذهن   | mind          |
> | 하나   | one       | یکی   | one           |
> | 평화   | peace     | آرامش | peace of mind |
> | 불     | fire      | آتش   |fire          |
> | 학교   | school    | مدرسه | school        |
>
> Furthermore, to highlight the proximity of semantically similar words, we calculated distances between corresponding Korean and Farsi words, juxtaposed against the average distance between the Korean word and a broader set of commonly-used Farsi words.
>
> Table 3: Distances between semantically similar words vs. average distances.
>
> |     ko     |   fa    |  distance   | average  distance |  $\Delta$   |
> | :--------: | :-----: | :---------: | :---------------: | :---------: |
> | 안녕하세요 |  آفتاب  | 8317.399414 |    9544.922852    | 1227.523438 |
> | 고마워하는 | خداحافظ | 15522.97461 |    17568.11328    | 2045.138672 |
> |     예     |  منون   | 4474.33252  |    8007.143066    | 3532.810547 |
> |   아니요   |   بله   | 10817.47852 |    11549.48438    | 732.0058594 |
> |    제발    | ببخشید  | 10472.74121 |    11903.8916     | 1431.150391 |
> |     큰     |  لطفاً   | 3131.273193 |    7231.745117    | 4100.471924 |
> |    작은    |  بزرگ   | 2274.627197 |    6987.947754    | 4713.320557 |
> |    친구    |  کوچک   | 3417.742676 |    7057.731934    | 3639.989258 |
> |    좋다    |  دوست   | 9683.666992 |    10981.41797    | 1297.750977 |
> |     집     |   عشق   | 3705.327393 |    7983.23877     | 4277.911377 |
> |    학교    |  خانه   | 2197.190186 |    7114.350586    |  4917.1604  |
> |     책     |  مدرسه  | 2706.901855 |    7249.868652    | 4542.966797 |
> |     펜     |  کتاب   | 10441.48047 |    11023.87305    | 582.3925781 |
> |    색상    |   قلم   | 8703.549805 |    11031.95801    | 2328.408203 |
> |     물     |   رنگ   | 4471.234375 |    7806.318848    | 3335.084473 |
> |    음식    |   آب    | 3287.777832 |    7130.495117    | 3842.717285 |
>
> Our findings indicate that words with parallel meanings tend to have reduced distances in vector space, validating the efficacy of multilingual machine translation even across script-divergent languages.
>
> > Q3: The paper mentions variations in the domain of the data utilized in the abstract, but it is not later investigated further. Does having different domain help or hinder the outcome? Are the results better because the model is seeing new English sentences and does not have that much to do with the source language?
>
> Our paper did indeed mention variations in the data domain in the abstract, and we apologize for not delving deeper into this in the main content. This was primarily due to constraints on paper length and our focus on other key aspects of the study.
>
> To directly address your query on domain variations: Yes, domain similarity between the training set and test set does play a decisive role in model performance. Table 1 presents this correlation: as the similarity between the two sets increases, the performance during fine-tuning also tends to rise.
>
> For a more specific illustration, we examined the fa language. We assessed its training set's similarity against the training sets of the language family (bn, hi) and the pseudo language family (tr, id, ko). The results, as displayed in Table 4, demonstrate that the auxiliary training data's similarity to the target language's training data is proportional to improved translation outcomes.
>
> Table 4: Similarity between language family and pseudo language family
>
> |      | bn                  | hi                  | tr                  | id                  | ko                  |
> | ---- | ------------------- | ------------------- | ------------------- | ------------------- | ------------------- |
> | fa   | 0.11492737485758579 | 0.11611198283623779 | 0.12014097928752948 | 0.12008474558327929 | 0.12465920288568932 |
>
> Importantly, our methodology primarily emphasizes enhancing the translation capability across bilingual pairs. This approach is not just about exposing the model to new English sentences; it's about fortifying its foundational bilingual translation competence. Our results suggest that the model's performance is influenced both by domain similarity and by the nuances of source languages. Our research underscores that a careful balance between these elements is pivotal for the optimal effectiveness of translation tasks.
>
> > Q4: If I understood correctly, you did not try to fine-tune each of the languages on all of the languages of the TED dataset?
>
> Indeed, we hadn't initially elaborated on that specific approach. Based on your suggestion, we fine-tuned the fa-en task across the entirety of the TED dataset. We subsequently used the resulting universal model as the foundation for further experiments. Here are the results:
>
> Table 5: BLEU Scores for the fa-en with Different Fine-Tuning Strategies
>
> |                   | **fa-en**   |
> | ----------------- | ---- |
> | Pseudo Family           | **36.9** |
> | Entire TED Dataset       | 36.2 |
> | TED Dataset + Pseudo Family | 36.7 |
>
> > Q5: In Figure 4, the similarity is always 1 if the language is itself. Would it still be 1, if the material would not be the same?
>
> Thanks for the question. We split the training set into two subsets of equal data size and recalculated the fisher information matrix for analysis. Of course, the similarity on different subsets will not be 1, but it is still much higher than other languages. We also test them on a different corpus, we calculate $S_{(t,a)} $ where $t$ is in the TED dataset and $t$ is in the OPUS dataset, the result shows in different materials the language itself still contains better similarity and our method has involved the information about language itself.
>
> > Q6: In 2.1 on lines 110 and 111 it said that the respective volumes of the dataset would be presented on Table 1. They are not presented there.
>
> We sincerely apologize for this oversight. We'll address this issue in the upcoming revision. Here is the missing table detailing the number of sentences for each language:
>
> Table 6: Number of Sentences for Each Language in the Dataset
> | language | sentences |
> | -------- | --------- |
> | **bg**       | 174,444    |
> | **sr**       | 136,898    |
> | **hr**       | 122,091    |
> | **uk**       | 108,495    |
> | **sk**       | 61,470     |
> | **be**       | 150,965    |
> | **fa**       | 18,798     |
> | **hi**       | 4,649      |
> | **bn**       | 4,509      |
> | **id**       | 87,406     |
> | **ms**       | 5,220      |
> | **kk**       | 3,317      |
> | **tr**       | 182,470    |
> | **de**       | 167,888    |
> | **ja**       | 204,090    |
> | **ko**       | 205,640    |
> | **sv**       | 56,647     |

---

### Official Review · Reviewer_yzL1 · 2023-08-04

**Soundness:** 3

**Excitement:**

3: Ambivalent: It has merits (e.g., it reports state-of-the-art results, the idea is nice), but there are key weaknesses (e.g., it describes incremental work), and it can significantly benefit from another round of revision. However, I won't object to accepting it if my co-reviewers champion it.

**Paper Topic And Main Contributions:**

This paper presents a way to select auxiliary languages to improve machine translation performance within a multilingual setting. Specifically, the paper proposes to compute similarity between a language pair t, a as a function of the Fisher Information Matrix of the parameters when translating into t and the Fisher Information Matrix when translating into a. Intuitively, the Fisher Matrix encapsulates the variance of the (log) derivative, so if two target languages overlap a lot in terms of which parameters have high variance, then information from the languages should help each other. The authors provide justification for the subset of parameters they choose based on an analysis, and also show that their method improves over several reasonable baselines (fine-tuning, language family-based, both together), but by less than a BLEU point on average.

**Questions For The Authors:**

- Is there any way to use the Fisher Information itself (e.g. through some distance function or similarity of gradients) instead of relying on this masking?
- Table 2 is very interesting, and shows some unintuitive relationships (e.g. fa with ko, id with de), any hypotheses about this?

**Reasons To Accept:**

- contribution towards the insight that the beneficial effect of auxiliary languages in multilingual training can be largely explained by the model parameters rather than external linguistic or typological constraints.

**Reasons To Reject:**

- the usage of a Fisher Mask and a similarity criterion based on this seems somewhat arbitrary, including choice of taking top 30% of parameters (was this tuned? why not another number).

**Reproducibility:**

3: Could reproduce the results with some difficulty. The settings of parameters are underspecified or subjectively determined; the training/evaluation data are not widely available.

**Reviewer Confidence:**

3: Pretty sure, but there's a chance I missed something. Although I have a good feel for this area in general, I did not carefully check the paper's details, e.g., the math, experimental design, or novelty.

---

> ### Author Rebuttal · Authors · 2023-08-29
>
> Thanks for the review.
>
> > Q1: The usage of a Fisher Mask and a similarity criterion based on this seems somewhat arbitrary, including choice of taking top 30% of parameters (was this tuned? why not another number).
>
> The decision to use the top 30% of parameters was made after extensive experimentation. We evaluated the model's performance on clustering pseudo language families using different thresholds. It was observed that the threshold of 30% yielded the best performance among the tested values, providing a good balance between capturing significant parameters and method efficiency.
>
> > Q2: Is there any way to use the Fisher Information itself (e.g. through some distance function or similarity of gradients) instead of relying on this masking?
>
> Thanks for emphasizing the thought of using Fisher Information itself. In our initial tests, we indeed explored various clustering strategies. We utilized fisher information, a feature intrinsic to neuron activation, to conduct these experiments. For distance measurement between language pairs, we applied MSE ($MSE = (F_t - F_a)^2/|F|$, where $F$ denotes the Fisher information matrix). While its performance was noteworthy, it was surpassed by the overlapping similarity on the fa-en task, as illustrated below:
>
> ********************
> |      | BLEU          |Auxilary       |
> | ------------- | -------- | --------- |
> | Baseline      | 25       |           |
> | MSE           | 36.3     | id tr de  |
> | Pesudo family | **36.9** | tr id ko  |
> ********************
>
> We will incorporate a dedicated section elaborating on the outcomes of various clustering techniques in the revised version.
>
> > Q3: Table 2 is very interesting, and shows some unintuitive relationships (e.g. fa with ko, id with de), any hypotheses about this?
>
> Thanks for pointing out this unusual outcome. It is truly unintuitive and absurd if we want to understand it from the perspective view of linguistics. However, once we feed those data to the pre-trained M2M model, they would be first represented by vectors containing semantic information. At the same time, in the word vector, words with similar meanings in different languages will have a closer distance, which can play a synergistic role in training. Therefore, take **fa** as an example. We measured the similarity using Euclidean distance between the training set of auxiliary languages and the training set of fa.
>
> To calculate the similarity:
> 1. Randomly select 100 data points, denoted as T, from the training set of auxiliary languages.
> 2. Randomly select 100 data points, denoted as S, from the training set.
> 3. For each data point in T, compute the cosine distance to all data points in S.
> 4. For each data point in T, average the cosine distances of its top 5 closest data points in S.
> 5. Finally, average the calculated similarities for all data points in T.
>
> The similarities are shown below:
> ********************
> | | Langauge Family| |Pseudo Family | | |
> | ---- | -------------------- | ------------------- | ------------------ | ------------------- | ------------------- |
> | |**bn-en**                      | **hi-en**                     | **tr-en**                    | **id-en**                     | **ko-en**                    |
> | **fa-en**   | 0.11492737485758579 | 0.11611198283623779 | 0.12014097928752948 | 0.12008474558327929 | 0.12465920288568932 |
> ********************
> **bn** and **hi** are in the language family and the rest languages in the table are auxiliary languages. It shows that the auxiliary language that has high similarity in data could lead to better performance. The observation is also in lined with **id**.
> ********************
> ||Language Family |Pseudo Family ||
> | ---- | ------------------- | ----------------- | ------------------ |
> | | **ms-en**                  | **bg-en**                | **de-en**                 |
> | **id-en**   | 0.11061211301192167 | 0.11564603335590569 | 0.11717251327650846 |
> ********************
> Therefore, our clustering results make sense from the perspective of model representation and data similarity. Languages with more similar data are more likely to be selected even if their similarity in linguistics is very low and better results can be obtained.

---

### Official Review · Reviewer_BkKF · 2023-08-07

**Soundness:** 3

**Excitement:**

4: Strong: This paper deepens the understanding of some phenomenon or lowers the barriers to an existing research direction.

**Paper Topic And Main Contributions:**

The paper detailed a simple approach of computing FIM, which in-turn was converted into activated neurons to represent each language for clustering.  The results, as summarized in table 2, seem to be quite reasonable, and experiments showed positive improvements.

Idea and experiment wise, the approach is simple and straightforward. Though some follow up details can be added to enrich the paper further, I think it is a good asset for the research community.

The authors claimed they will release code & data.

**Reasons To Accept:**

Idea and experiment wise, the approach is simple and straightforward. Though some follow up details can be added to enrich the paper further, I think it is a good asset for the research community.

The authors claimed they will release code & data.

**Reasons To Reject:**

The $S(t, a)$ is using the activated neurons and binarized them into 0 or 1.  This seems to be adhoc, and if the authors can provide using floats of these activated neurons for similarly/distance computing for clustering, the compare and analysis will be great.

**Reproducibility:**

4: Could mostly reproduce the results, but there may be some variation because of sample variance or minor variations in their interpretation of the protocol or method.

**Reviewer Confidence:**

4: Quite sure. I tried to check the important points carefully. It's unlikely, though conceivable, that I missed something that should affect my ratings.

---

> ### Author Rebuttal · Authors · 2023-08-29
>
> Thanks for the review.
>
> > Q1: The $S_{(t,a)}$ is using the activated neurons and binarized them into 0 or 1. This seems to be adhoc, and if the authors can provide using floats of these activated neurons for similarly/distance computing for clustering, the compare and analysis will be great.
>
> Thanks for highlighting the comparison with other distance algorithms. In our initial tests, we indeed explored various clustering strategies. We utilized fisher information, a feature intrinsic to neuron activation, to conduct these experiments. For distance measurement between language pairs, we applied MSE ($MSE = (F_t - F_a)^2/|F|$, where $F$ denotes the Fisher information matrix). While its performance was noteworthy, it was surpassed by the overlapping similarity on the fa-en task, as illustrated below:
>
> ********************
> |      | BLEU          |Auxilary       |
> | ------------- | -------- | --------- |
> | Baseline      | 25       |           |
> | MSE           | 36.3     | id tr de  |
> | Pesudo family | **36.9** | tr id ko  |
> ********************
>
> We will incorporate a dedicated section elaborating on the outcomes of various clustering techniques in the revised version.

---

### Meta-Review · Area_Chair_f8Fs · 2023-09-19

**Recommendation:** 4

**Metareview:**

This paper presents a new method for selecting auxiliary languages to improve machine translation. Instead of using traditional language families, they compute similarity between each pair of languages with Fisher Information Matrix, so that languages will be considered similar if they overlap a lot in terms of which parameters have high variance, and use this to compute pseudo-families based on the empirical data. The technique seems like it should be also applicable for other situations that would benefit from measuring language similarity.

The reviewers appreciated that the apprach is straightforward and effective.

---

### Decision · Program_Chairs · 2023-10-07

**Decision:**

Accept-Main

**Comment:**

This paper presents a new method for selecting auxiliary languages to improve machine translation. Instead of using traditional language families, they compute similarity between each pair of languages with Fisher Information Matrix, so that languages will be considered similar if they overlap a lot in terms of which parameters have high variance, and use this to compute pseudo-families based on the empirical data. The technique seems like it should be also applicable for other situations that would benefit from measuring language similarity.

The reviewers appreciated that the apprach is straightforward and effective.